# The Effect of the NorGeP–NH on Quality of Life and Drug Prescriptions in Norwegian Nursing Homes: A Randomized Controlled Trial

**DOI:** 10.3390/pharmacy10010032

**Published:** 2022-02-16

**Authors:** Enrico Callegari, Jurate Šaltytė Benth, Geir Selbæk, Cato Grønnerød, Sverre Bergh

**Affiliations:** 1Østfold Hospital Trust, 1714 Grålum, Norway; cato.gronnerod@psykomatikk.no; 2Faculty of Medicine, University of Oslo, 0372 Oslo, Norway; geir.selbaek@aldringoghelse.no; 3Institute of Clinical Medicine, Campus Ahus, University of Oslo, 0372 Oslo, Norway; jurate.saltyte-benth@medisin.uio.no; 4Health Services Research Unit, Akershus University Hospital, 1478 Lørenskog, Norway; 5Research Centre for Age-Related Functional Decline and Diseases, Innlandet Hospital Trust, 2312 Ottestad, Norway; sverre.bergh@sykehuset-innlandet.no; 6Norwegian National Centre for Ageing and Health, Vestfold Hospital Trust, 3103 Tønsberg, Norway; 7Department of Geriatric Medicine, Oslo University Hospital, 0372 Oslo, Norway; 8Department of Psychology, Faculty of Social Sciences, University of Oslo, 0317 Oslo, Norway

**Keywords:** psychotropic polypharmacy, structured drug review, nursing homes

## Abstract

Background: The effect of the Norwegian General Practice–Nursing Home (NorGeP–NH) criteria has never been tested on clinical outcomes in nursing home (NH) residents. We performed a cluster-randomized trial in Norwegian NHs and tested the effect of NorGeP–NH on QoL (primary outcome), medication prescriptions, and physical and mental health (secondary outcomes) for the enrolled residents; Methods: Fourteen NHs were randomized into intervention NHs (iNHs) and control NHs (cNHs). After baseline data collection, physicians performed NorGeP–NH on the enrolled residents. We assessed the difference between cNHs and iNHs in the change in primary outcome from baseline to 12 weeks and secondary outcomes from baseline to eight and 12 weeks by linear mixed models; Results: One hundred and eight residents (13 lost to follow-up) and 109 residents (nine lost to follow-up) were randomized to iNHs and cNHs, respectively. Difference in change in QoL at 12 weeks between cNHs and iNHs was not statistically significant (mean (95% CI)): −1.51 (−3.30; 0.28), *p* = 0.101). We found no significant change in drug prescriptions over time. Difference in depression scores between cNHs and iNHs was statistically significant after 12 weeks. Conclusions: Our intervention did not affect QoL or drug prescriptions, but reduced depression scores in the iNHs. NorGeP–NH may be a useful tool, but its effect on clinical outcomes may be scarce in NH residents. Further studies about the effectiveness of NorGeP–NH in other healthcare contexts and settings are recommended.

## 1. Introduction

It is well established that polypharmacy, often defined as the use of more than five concomitant drugs [1], is prevalent in nursing homes (NHs) and is associated with frailty, hospitalization, cognitive and physical impairment, falls, and mortality [2,3].

In the past years, several explicit lists, such as Beers Criteria [4,5], START/STOPP [6], EU (7)-PIM [7], the PRISCUS list [8], and NorGeP [9], have been introduced to identify potentially inappropriate medications (PIM) in older adults. Medications may be inappropriate when their potential harm exceeds their benefit [10]. Over 30 PIM lists have been published between 1991 and 2017 aiming to identify the complexity of drug therapy in older people, but these lists have wide variability in what is considered a PIM [11]. This variability can cause a big discrepancy in the detection of PIMs according to which list a clinician uses [12,13,14].

Several authors have approached this complexity by developing multifaceted interventions to avoid polypharmacy in older people, where medication review in an important aspect [15]. However, RCTs aiming to evaluate the effect of medication reviews on clinical, drug-related, and organizational outcomes are heterogeneous, do not always use standardized clinical outcomes, and lead to opposite or not always robust conclusions [16,17].

A recent meta-analysis of national and international studies showed that psychotropic polypharmacy, defined as the use of two-or-more or three-or-more psychotropic drugs (PTDs), is common in NH residents with dementia [18]. PTDs, such as antidepressants, antipsychotics, anxiolytics, sedatives, hypnotics, and antidementia drugs, may be used not only to treat primary psychiatric disorders, but also to mitigate neuropsychiatric symptoms (NPSs) associated with dementia. NPSs can be delusions, hallucinations, agitation, aggression, depression, anxiety, euphoria, apathy, disinhibition, irritability, aberrant motor behaviors, nighttime behaviors, and changes in appetite. According to national guidelines, clinicians should carefully assess the appropriateness of PTD prescribing in older people with dementia [19]. However, between 2000 and 2016, antipsychotics were the only PTD with reduced prescription in NH residents in Norway [20]. Despite a large variation in PTD prescriptions between different NHs, PTDs still cause about one-third of the detected drug-related problems among NH residents [21].

Although medication reviews can be helpful to reduce PTD prescriptions, it might not be clear if this reduction is clinically beneficial [22]. A recent cluster-randomized trial conducted in the Netherlands, for example, showed no reduction in PTD prescriptions and NPS occurrence despite a multidisciplinary intervention [23]. Similarly, another cluster-randomized trial conducted in German NHs, did not affect the prescription of potentially inappropriate medication or neuroleptic drugs, despite conducting a complex intervention [24]. Some studies have shown that when a medication review is performed by non-prescribers, such as pharmacists or external teams without the same knowledge of a resident as the primary physician, it might lead to a discrepancy between identified PIMs or suggested medication changes from pharmacists and the actual changes performed by a physician [22,25,26]. Therefore, it is important that healthcare personnel in NHs, including NH physicians, have sufficient knowledge about the correct use of PTDs in NH residents with psychiatric symptoms, and come to a joint decision, through team collaboration, about the necessary medication prescription after a medication review.

The Norwegian General Practice – Nursing Home (NorGeP–NH) is a list of criteria used to perform a medication review [27]. It is divided into three groups: single-substance criteria, drug–drug combination criteria, and criteria where regular consideration of “de-prescribing” is of utmost importance in an NH population [27]. This list has previously been used to identify PIMs, but, as far as we are aware, it has never been tested in a “real-world” clinical setting, as a recent systematic review presented [28].

Self-perceived Quality of life (QoL) embraces many aspects of a person’s physical and emotional health and gives a broader idea of the level of disease burden a person experiences. Measuring QoL in people with dementia can be challenging, as their level of insight might decrease as dementia worsens [29]. However, an observation-based scale such as QUALID has shown to be reliable and associated with depression, level of functioning, degree of dementia, agitation, and psychosis [30]. QoL, measured with either self-based or proxy-based tools, is associated with several clinical factors, including polypharmacy [31]. Despite a large number of assessment tools listing PIMs, only a few studies have presented the effect of PIM assessment tools on different persons’ outcomes, and even fewer studies have explored a possible association between a specific PIM assessment tool and QoL [28]. In fact, only two PIM assessment tools have been explored and were found positively associated with an improvement in QoL [32,33].

The main objective of our study was to examine whether QoL (primary outcome) in NH residents could be influenced by exposing NH physicians to an educational program about NorGeP–NH, after receiving a lecture on psychotropic drug use in older adults, and requesting them to perform a structured medication chart review with NorGeP–NH. As for secondary outcomes, we examined whether the same intervention influenced PTD and total medication prescriptions, cognitive function, NPS, physical health status, and functioning in daily living in the same group of residents.

## 2. Materials and Methods

### 2.1. Trial Design

We performed a cluster-randomized trial in 14 NHs, with a total of 42 wards, distributed in eight municipalities in Østfold county, Norway, between November 2018 and June 2019. NHs were treated as clusters, as the intervention was at staff/physician level and not at resident level. Primary and secondary outcomes were at resident level. The NHs were cluster-randomized into two groups, and the NHs were given the name of intervention NHs (iNHs) and control NHs (cNHs). Allocation was not revealed to the NH personnel until after completion of baseline data collection in order to minimize detection bias at baseline. Many of the chosen assessment tools needed to be administered by nurses/authorized social workers who knew the participants well and who had observed the participants over time. Thus, it was not possible to blind data collectors after the intervention was delivered. The report of this trial follows the recommendations of CONSORT (Consolidated Standards of Reporting Trials) guidelines and CONSORT extension to cluster-randomized trials [34]. The described intervention follows the TIDieR criteria [35].

Every participant gave written informed consent to be included in the trial. The capacity to consent was evaluated by a clinical examination performed first by the NH physician, and confirmed by the first author, to detect the participant’s ability to understand and weigh the given information, reason, and give an explicit choice. In case of doubt, clinicians could use the Aid to Capacity Evaluation (ACE) form [36]. If participants had reduced capacity to consent, a written informed consent was obtained from the participant’s next of kin. The Regional Committee for Medical and Health Research Ethics (2017/2171 REK south-east D) approved the trial. The study was registered on 6 November 2018, on clinicaltrials.gov (accessed on 9 January 2022) (NCT03736577).

### 2.2. Participants

Before inclusion, all 19 municipalities in the district served by the regional Østfold Hospital, with a total of 34 long-term care NHs, received information about the study protocol and were invited to participate. Those responsible for healthcare services in every municipality decided which nursing home(s) could participate. Once the participating NHs were determined, the responsible NH physicians were informed about eligibility criteria to include the NH residents in this study. Eligibility criteria were (a) NH resident and (b) expected to live in the NH for more than 12 weeks. Exclusion criteria were (a) terminal disease, (b) severe somatic or psychiatric disease where the resident was too debilitated or not able to cooperate or where the examination would cause too great of a psychological and physical burden (i.e., severe psychotic state), and (c) the physician had performed a structured drug review for the participant within three months prior to inclusion. NH physicians were thoroughly informed about these criteria and were responsible for assessing eligibility.

Prior to baseline data collection and randomization, the healthcare personnel from both iNHs and cNHs participated in a three-hour lecture on dementia and dementia-related neuropsychiatric symptoms, delirium, depression, anxiety, and psychosis in older people. In addition, we asked each participating nursing home to dedicate one or two NH personnel per ward to collect data. The data collectors were nurses or authorized social workers, and they participated in a three-hour lecture to learn how to use validated assessment tools. The assessment tools were either interview-based or proxy-based, and they are described later. The first author gave both lectures. Clinical data about the residents in both iNHs and cNHs were collected eight and 12 weeks after baseline data collection.

### 2.3. Intervention

The intervention was an *educational intervention* on nursing home physicians, followed by a *drug chart review* of the participant’s medications, and included the following steps:

(1) Physicians in the iNHs attended a three-hour lecture including the following subjects:-principles of pharmacology in older people;-the use of PTDs in older people;-how to conduct a drug chart review with the Norwegian General Practice–NH (NorGeP–NH) criteria [27].

This lecture was held by a psychiatrist (first author) after baseline data were collected. The lecture was held in the nursing home where the physicians worked. It was held face-to-face and included an electronic presentation as supportive material. The physicians who attended the lecture were given a copy of the electronic presentation after the lecture, and they received a laminated NorGeP–NH list to use in the following step.

(2) Within a two-week period after the lecture, physicians in the iNHs performed a drug chart review according to NorGeP–NH. Physicians were allowed to consult a psychiatrist (first author) in case they needed to discuss choices made during a review, but the final decision about medication changes was the physician’s responsibility.

### 2.4. Control Group

The physicians and healthcare personnel in the cNHs were asked to follow-up residents as usual. If medication changes were necessary, physicians could do so, but without using a structured drug review chart during the follow-up period. After the last assessment at 12 weeks, as a courtesy to the physicians in the cNHs they were given the same lecture as described in (1).

### 2.5. Collected Data and Outcomes

The primary outcome was the difference between cNHs and iNHs in change in quality of life (QoL) assessed with the Quality of Life in Late-Stage Dementia (QUALID) scale [30,37], from baseline to 12-week follow-up. The secondary outcomes were the difference between cNHs and iNHs in change from baseline to 8–12 weeks in the number of drugs prescribed daily, the number of prescribed pro re nata (PRN) drugs, the prescription of psychotropic drugs categories (antidepressants, antipsychotics, anxiolytics, sedatives/hypnotics, and antidementia drugs), and in clinical scores measuring the level of depression, cognitive function, neuropsychiatric symptoms, physical health status, and functioning in daily living. Table 1 reports the instruments used to collect the data. We also collected demographic data and nursing home characteristics for each participating resident (Table 2 in Results section).

### 2.6. Sample Size

In a previous Norwegian study, people admitted to an NH had a QUALID score of 20.0 (SD 7.2) [51]. When the study was designed, to the best of our knowledge, there were no previous randomized controlled studies using QUALID score as a primary outcome. Thus, to be sure that any possible change caused by our intervention was clinically relevant, we chose to define a change from baseline to 12-week follow-up in QUALID score of 33% as of clinical importance prior to power calculations. With an 80% power and 0.05 significance level, and assuming SD 7.2 in both groups at baseline and follow-up, 39 residents needed to be included in each allocation group to detect a 33% difference between iNHs and cNHs in change from baseline to 12-week follow-up in QUALID score. In Norwegian NHs, about one out of four residents die every year [52]. Thus, we estimated a 6–7% drop-out rate due to death within a 12-week period. Rounding up the drop-out rate to 10%, 43 residents had to be included in each allocation group. Assuming 10 participating NHs and a cluster effect on NH level of about 5%, the final number of residents was estimated to be 60 in each group. Because of uncertainty about how many NHs would decide to participate in the study, we aimed to include about 100 residents in each allocation group.

### 2.7. Randomization

An independent statistician allocated the participating NHs into two arms by performing a stratified randomization using a computer-generated algorithm. To avoid contamination bias, every NH was treated as a cluster. Each NH was under the care of one physician or group of physicians who worked together and only in that NH. All the participating NHs were stratified into four groups. Stratification was performed according to the number of participants the personnel in each NH were able to include and follow up. The allocation results were kept in a digital safe, hidden from NH physicians responsible for enrolling participants. NH physicians were asked to assess each resident in the participating NH for eligibility, and they were responsible for enrolling participants. One of the authors (EC) verified the eligibility criteria by discussing them with the NH physician/NH personnel and verified the participants’ capacity to consent. If an NH had limited resources to follow up participants, NH leaders and physicians were asked, before inclusion and allocation, to determine how many residents they could possibly enroll and follow up. In this case, the predetermined number of residents was selected by drawing lots. This process was performed by EC in the presence of at least one healthcare personnel from the selected NH.

Once NHs were allocated and residents were enrolled, a random-number generator was used to determine which allocation group was given the intervention. The result of this process was also kept hidden from NH physicians and healthcare personnel until after baseline data were collected. Once baseline data were collected, the first author informed the physicians who were working in the intervention NHs and carried out the intervention together with them.

### 2.8. Statistical Methods

The statistical analyses were performed by using SPSS© v27 and SAS© v9.4. Baseline characteristics are presented as means and standard deviations (SDs) for continuous variables and frequencies and percentages for categorical variables. We present the total amount of prescribed drugs as means (SDs) both for daily prescriptions and for pro re nata (PRN) drugs. PTD prescriptions are presented as frequencies and percentages at each assessment point. For the primary and secondary analyses, we included participants who had data available at baseline. To assess the difference between iNHs and cNHs in the change in primary and secondary continuous outcomes, we estimated linear mixed models with fixed effects for time, allocation group, and interaction between these two. To assess the difference in change for categorical outcomes, we estimated generalized linear mixed models with the same fixed effects. All models contained random effects for NHs to adjust the estimates for cluster effect at the NH level, which was non-negligible according to the intra-class correlation coefficient. For continuous outcomes, the results were presented as mean change within allocation groups and mean difference in change between the groups with corresponding 95% confidence interval (CI) and *p*-value. For categorical outcomes, the results were presented as odds for change within the allocation group as well as odds for differences in change between groups with 95% CI and *p*-values. We set the level of significance at 5%.

## 3. Results

Figure 1 shows the flow diagram of the trial. Two hundred and seventeen residents were included at baseline between November 2018 and March 2019. Six hundred and three residents from 15 NHs in the nine municipalities that agreed to participate in the trial were recruited and assessed for eligibility. Among these, 437 met inclusion criteria. One NH in one municipality, which had originally agreed to participate with 14 residents, withdrew from the trial during eligibility assessment due to a lack of local NH resources. Fifty residents declined to participate, 161 residents were excluded by drawing lots because some NHs could not include more than a predetermined number of residents (see Section 2.7), six residents died right before baseline assessment, and one resident moved from the NH right before baseline assessment. Two residents were excluded for violation of protocol, as the NH never returned the assessment documentation. Sixteen NH physicians were involved in the trial, seven working in the cNHs and nine working in the iNHs.

Table 2 reports demographics, NH characteristics, and clinical scores at baseline. Residents were on average (SD) 84.6 (9.4) and 83.3 (8.0) years old in the cNHs and iNHs, respectively. Most residents in the control group lived in regular units (56.9%), while most residents in the intervention group lived in special care units (59.3%). The two groups had a comparable number of residents per unit (15.07 in cNHs vs. 13.15 in iNHs), number of staff members per unit during the day shift (4.73 in cNHs vs. 4.61 in iNHs), and physicians worked on average 0.88 more hours in cNHs compared to iNHs (6.43 h in cNHs vs. 5.55 h in iNHs). According to CDR, most participants had either mild cognitive impairment (7.8% in cNHs and 7.7% in iNHs) or dementia (89.3% in cNHs and 92.3% in iNHs). The average (SD) number of prescribed daily drugs was 6.92 (3.49) for participants living in the cNHs and 7.55 (3.04) for participants living in the iNHs. The average number (SD) of prescribed pro re nata (PRN) drugs was 4.04 (2.74) for the cNHs and 4.72 (2.89) for the iNHs. For some of the assessment scores, such as MoCA, CSDD, QUALID or TUG, there was a considerable amount of missing data. This aspect is discussed later.

Results from the primary analysis, assessing the difference in change in QoL, are presented in Table 3. We found no statistically significant difference between cNHs and iNHs in change in QoL from baseline to 12-week follow-up. However, while the QUALID score remained stable in the iNHs, we found a statistically significant increase in QUALID score (higher QUALID score indicates lower QoL) in the cNHs (*p* = 0.013).

Results from the analyses of secondary outcomes (see paragraph “Outcomes” and Table 1 for details) are presented in Table 4 for clinical measures and Table 5 for prescribed drugs. Compared to the control group, residents in the iNHs had a significantly larger reduction in CSDD score from BL to week 12 (mean difference in change (95% CI) −2.59 (−3.95; −1.23), *p* < 0.001). We found no statistically significant difference between the two groups in change in the prescription of PTD categories (antidepressants, antipsychotics, anxiolytics, sedatives/hypnotics, and antidementia drugs treated as groups).

Further, compared to the control group, participants in the iNHs had a statistically significant reduction in GAI score from BL to week 8 (−1.69 (−3.37; −0.01), *p* = 0.049), and a statistically significant reduction in the total amount of prescribed daily medications from BL to week 8 (−0.41 (−0.75; −0.06), *p* = 0.023). Residents in the iNHs, compared to residents in cNHs, had a significantly larger reduction in the odds of having a lower CDR score from baseline to week 8 (*p* = 0.007), but no significant difference in reduction from baseline to week 12.

## 4. Discussion

### 4.1. Brief Synopsis of Key Findings

This NH trial examined how an educational program on psychopharmacology and the use of NorGeP–NH in a real-world setting influenced QoL, other clinical outcomes, and medication prescriptions among NH residents. Our trial did not show any significant difference in change in QoL scores between iNHs and cNHs from BL to 12-week follow-up. Even though we found a statistically significant reduction in QoL among cNHs residents from BL to 12-week follow-up, this reduction was not relevant from a clinical perspective, according to what we assumed to be a clinically important reduction (33%). Our trial showed that the intervention did not reduce the total amount of daily prescribed drugs in the iNHs compared to cNHs after 12 weeks. However, there was a significant, yet temporary, reduction of the total amount of daily prescribed drugs from BL to 8 weeks, in the iNHs. Our results also showed that the depression score was significantly lower in the intervention group compared to the control group, at 12-week follow-up. Our trial did not show any significant difference between cNHs and iNHs in change for PTD category prescriptions.

### 4.2. Strengths and Limitations

A strength of this study is the fact that participants were assessed by healthcare personnel who worked in the NHs where the participants lived and who had good knowledge of the participants’ clinical history. We also chose to focus on a “real-world” intervention performed by the same physicians treating the participants in the usual care setting, and not by conducting the intervention by external personnel, which has been previously discussed and has shown a possible lower adherence to suggested medication changes [22,25,26].

Another strength of our study is that NH personnel performed clinical evaluations with validated tools, commonly used both in Norway and internationally in NHs. This makes it easier to compare our results with other studies. In addition, only one investigator (first author) directed the intervention and follow-up assessments, and by having direct contact with every participating NH, the possibility of missing data in the dataset was minimized.

This study has several limitations. Participants were cluster-randomized to minimize within-NH contamination bias. Despite cluster-randomization, the intervention and control groups may have had differences that impacted the results. For example, at baseline, a higher number of participants in the iNHs lived in special care units. This might reduce the potential beneficial effects of the performed intervention due to a higher level of morbidity in residents admitted to special care units, and, as a consequence of that, a lower QoL over time. Further, the short duration of the trial might not have been long enough to assess the long-term effects of the intervention. However, due to a high mortality rate in NHs [52], a shorter trial duration may reduce the number of people dropping out of the study.

Some assessment scores at BL, such as MoCA, QUALID, CSDD, GAI, or TUG, had several missing data. This may cause uncertainty when comparing the groups. We do not have an explanation for the reason why there were many missing data for QUALID or CSDD, as they are proxy-based assessment tools. For MoCA, GAI, and TUG, a possible explanation is that they require direct cooperation of the residents, which may have been difficult to achieve due to severe cognitive or physical impairment.

The data collectors were nursing home personnel, and they were blinded only during baseline data collection. This is a possible source of detection bias, as the assessors may have been influenced by the knowledge of randomization. However, most of the proxy-based assessment tools needed a deep knowledge of the participants and an observation time that lasted several days or weeks before an assessment was performed. Therefore, it would also be problematic to have the participants assessed by external, fully blinded personnel. Further, we did not analyze inter-rater reliability, and this may have led to bias in the data collection process, due to possible differences between data collectors. We did not collect precise data about how many nurses/authorized social workers participated in the data collection process, as this was decided internally in each nursing home ward. However, 42 wards participated in the study, and each ward had at least one or two data collectors. A large number of data collectors may also reduce a possible skewing of the distribution in the collected data.

Finally, we did not assess potential economic consequences of the intervention, which might be important to support such educational interventions in the future.

### 4.3. Considerations and Comparison with Relevant Studies

Our trial did not show a significant difference between the two groups in change in QoL. It is possible that our educational intervention, which focused on medication review and was only performed once, was not enough to improve QoL in the short term. In fact, a multicomponent, long-lasting intervention conducted in Norway showed no change in QoL during the first four months of intervention, but it showed an improvement nine months later [53]. However, our results are still in line with a Cochrane review conducted in 2016. That review analyzed the effect of different interventions to optimize drug prescriptions in NH residents and found no strong evidence showing intervention efficacy on resident-related outcomes, such as hospital admissions, mortality, adverse drug events, or QoL [16]. It is still important to note that even though we found no significant difference between the two groups in change in QoL, QoL did not worsen in the intervention group, while there was a significant reduction in QoL for participants in the cNHs. Indeed, the intervention might have prevented a possible worsening in QoL for the iNHs. However, the worsening in QUALID score we found in the control group may not be clinically relevant, as the mean (SD) score worsened from 21.31 (6.72) to 22.74 (7.64).

This trial showed a significant reduction in the total amount of prescribed drugs eight weeks after intervention, but not after 12 weeks. A possible explanation is that some drugs might have been reintroduced after a temporary discontinuation. However, we have not analyzed changes for all types of medication, as we only examined drug categories in the N05-/N06-ATC groups in detail. Our results seem to find partial support in previous studies. A recent systematic review showed that medication reviews can improve the appropriateness of drug prescription [17], which in some cases requires drug discontinuation. However, our trial may show that the effect of a medication review on the total medication amount is only temporary. The reduction in the total amount of drugs prescribed daily for older residents may have a beneficial effect on several clinical factors, such as reduction of frailty, improvement in cognitive function, or lower risk of falls [2]. However, among the clinical outcomes examined in our trial, only depression improved after 12 weeks. On the one hand, depression is an important risk factor of polypharmacy and excessive polypharmacy [54]. On the other hand, several somatic prescription medications, such as antihypertensives, proton pump inhibitors, or analgesics, are known to possibly cause depression as an adverse outcome [55].

Residents living in iNHs showed an increase in CDR score during follow-up. A possible explanation might be that more participants in the iNHs lived in special care units, often offered to people with moderate to severe dementia symptoms and poor prognosis. This difference might also explain why at BL residents living in iNHs presented more severe cognitive symptoms measured by CDR and MoCA as well as more severe NPS measured by NPI-NH.

Even though a recent systematic review and meta-analysis showed that focused psychotropic medication review is effective in reducing PTD prescriptions in NH residents [22], our intervention used a more general drug chart review tool which is not PTD-specific, and this might explain why we did not find a significant difference in change in PTD prescriptions between the two groups. NorGeP–NH is described in a recent review as limitative, as it does not include possible inappropriate medications for specific comorbidities [56]. However, a newly performed multidisciplinary, long-term NH cluster-randomized intervention, using other drug review lists, also failed to show an effect in reducing PTD prescriptions [23].

## 5. Conclusions

Our intervention on the use of NorGeP–NH and on the correct use of PTD in older NH residents did not have an effect on QoL or PTD prescriptions in the short term. However, our intervention still showed a positive, yet temporary, effect on the total drug load residents received and on the level of depression. NorGeP–NH may still have value in clinical practice, even if the evidence of its beneficial clinical effects may be scarce. Future research on the use of NorGeP–NH and other medication review tools should be performed in NHs and in other clinical settings to assess their real effectiveness on medication prescriptions and on the overall health status of older adults.

## Figures and Tables

**Figure 1 pharmacy-10-00032-f001:**
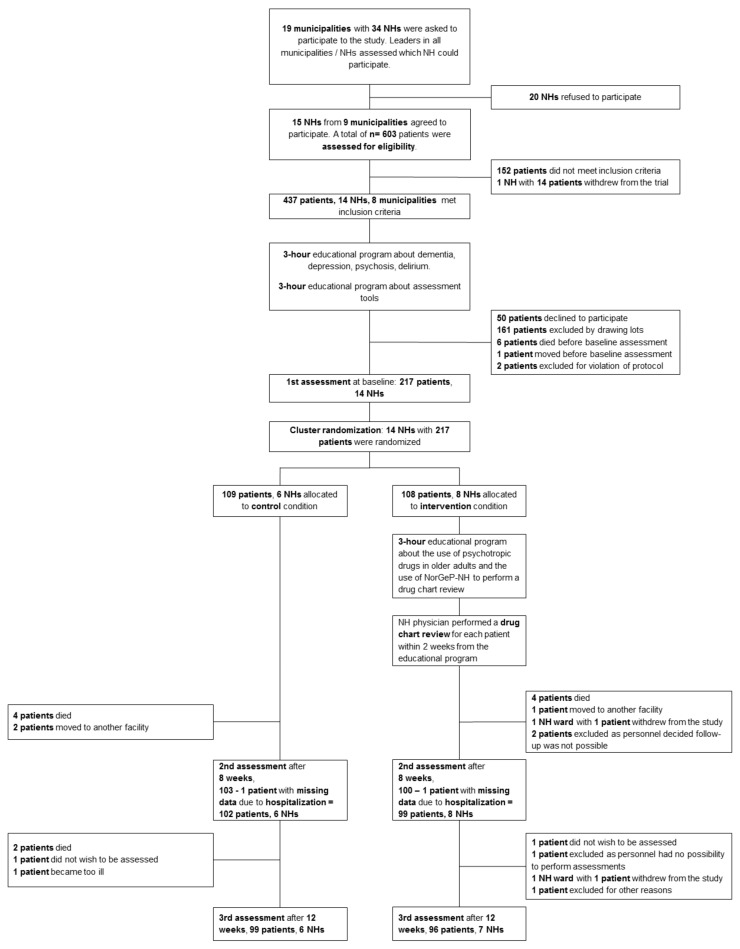
Flowchart of the trial. NH, nursing home.

**Table 1 pharmacy-10-00032-t001:** Structured interviews and checklists used to collect data ^a^.

Clinical Feature	Assessment Tools	Method of Collection	Ranging Score	Comments
Cognitive function	Montreal Cognitive Assessment (MoCA)	Interview	0–30	A higher score indicates better cognitive function [38].
	Clinical Dementia Rating (CDR) scale	Proxy-based	0–3	Total score is calculated using a complex algorithm. CDR = 0: no dementia; CDR = 0.5, 1, 2, or 3 indicates questionable, mild, moderate, or severe cognitive impairment [39].
Neuropsychiatric symptoms	Neuropsychiatric Inventory 12-item Nursing Home Version (NPI-NH) ^b^	Proxy-based	0–144	Single-item score is calculated by multiplying severity (score 1–3) by frequency (score 1–4). Total score is the sum of all the single-item scores [40,41,42]. We calculated the NPI-NH subsyndrome scores for agitation, psychosis and affective symptoms ^b^.
	Cornell Scale for Depression in Dementia (CSDD)	Proxy-based	0–38	Total score is calculated by summing up 19 single-item scores. Each single item can be scored 0, 1 or 2 (symptom not present, moderate or periodically present, severe). A higher score indicates more severe symptoms [43].
	Montgomery and Asberg Depression Rating Scale (MADRS)	Interview	0–60	Total score is calculated by summing up 10 single-items scores (0–6). A higher score indicates more severe symptoms [44].
	Geriatric Anxiety Inventory (GAI)	Interview	0–20	A 20-item self-report or nurse-administered scale. A higher score indicates more anxiety-related symptoms [45].
Medication	Anatomic Therapeutic Chemical (ATC) classification system	Medication chart in resident’s journal	N/A	We calculated the total amount of daily prescribed drugs, and the total amount of prescribed pro re nata (PRN) drugs. We collected data on prescribed psychotropic drugs, and we grouped them as antipsychotics (N05A except lithium), antidepressants (N06A), anxiolytics (N05B), hypnotic/sedatives (N05C), and anti-dementia medication (N06D).
Physical health status	General Medical Health Rating (GMHR) scale	Proxy-based	Excellent, good, fair, poor	Used to assess the general medical health status of each participant, according to the amount of stable/unstable medical conditions, the number of prescribed drugs and the general clinical condition [46].
	Charlson Comorbidity Index	N/A	0–30	A scale divided into 18 items/groups of diseases. Each item is scored yes/no, assuming the value of 1/0. An algorithm calculates the total score. Higher values indicate a higher level of comorbidity [47].
	Timed “Up and Go” test (TUG)	N/A	N/A	It measures the ability to stand up from a sitting position, walk a predefined distance, and sit down again. The score is in seconds and calculated as the average of two performances [48].
Functioning in daily living and quality of life (QoL)	Physical Self-Maintenance Scale (PSMS)	Proxy-based	1–6	A 6-item scale to measure the level of functioning. Each item is scored 1 only if there is no decline. A higher score indicates a higher level of functioning [49].
	Quality of Life in Late-Stage Dementia scale (QUALID)	Proxy-based	11–55	A 11-item assessment scale, where lower scores indicate a higher QoL [30,37].

^a^ Data were collected by nurses/authorized social workers. ^b^ A previous principal component analysis identified the NPI-NH subsyndromes: NPI-NH agitation (agitation/aggression, disinhibition, and irritability), NPI-NH psychosis (delusions and hallucinations), and NPI-NH affective (depression and anxiety) [50].

**Table 2 pharmacy-10-00032-t002:** Demographics, nursing home characteristics, and clinical scores at baseline.

	Control NHs (*n* = 109) ^a^	Intervention NHs (*n* = 108) ^a^
Age		
Mean (SD)	84.57 (9.43)	83.33 (7.97)
Gender		
Female, *n* (%)	78 (71.6)	61 (56.5)
Type of unit, *n* (%)		
Regular ^b^	62 (56.9)	44 (40.7)
Special care ^c^	33 (30.3)	64 (59.3)
Other	14 (12.8)	0 (0)
Number of residents per unit		
Mean (SD)	15.07 (4.41)	13.15 (3.97)
Number of staff members per unit on day shift		
Mean (SD)	4.73 (1.80)	4.61 (1.79)
Physician hours per week		
Mean (SD)	6.43 (1.68)	5.55 (3.52)
CDR, *n* (%)	*n* = 103	*n* = 104
0–no dementia	3 (2.9)	0 (0)
0.5–questionable cognitive impairment	8 (7.8)	8 (7.7)
1.0–mild cognitive impairment	30 (29.1)	20 (19.2)
2.0–moderate cognitive impairment	28 (27.2)	32 (30.8)
3.0–severe cognitive impairment	34 (33)	44 (42.3)
Charlson Comorbidity Index	*n* = 108	*n* = 101
Mean (SD)	2,54 (1.96)	2.57 (1.68)
CSDD	*n* = 94	*n* = 87
Mean (SD)	6.50 (5.84)	7.46 (5.99)
MADRS	*n* = 78	*n* = 45
Mean (SD)	9.03 (7.80)	7.47 (6.67)
GAI	*n* = 81	*n* = 56
Mean (SD)	5.58 (5.70)	5.0 (5.32)
GMHR, *n* (%)	*n* = 106	*n* = 99
Poor	0 (0)	11 (11.1)
Fair	44 (41.5)	50 (50.5)
Good	37 (34.9)	19 (19.2)
Excellent	25 (23.6)	19 (19.2)
MoCA	*n* = 79	*n* = 73
Mean (SD)	10.66 (6.97)	7.08 (6.44)
NPI-Total score	*n* = 107	*n* = 104
Mean (SD)	17.10 (19.10)	21.92 (21.30)
NPI-Caregiver	*n* = 107	*n* = 104
Mean (SD)	6.92 (8.50)	9.48 (10.49)
NPI-Affective ^d^	*n* = 107	*n* = 101
Mean (SD)	3.58 (5.46)	4.15 (5.42)
NPI-Psychosis ^d^	*n* = 101	*n* = 102
Mean (SD)	1.93 (3.72)	3.51 (4.73)
NPI-Agitation ^d^	*n* = 107	*n* = 104
Mean (SD)	5.26 (8.38)	8.20 (9.48)
PSMS		
Mean (SD)	1.06 (1.31)	1.16 (1.29)
QUALID	*n* = 97	*n* = 106
Mean (SD)	21.31 (6.72)	23.27 (8.03)
TUG	*n* = 40	*n* = 36
Mean (SD)	26.81 (16.67)	27.52 (20.36)
Number of daily medications		
Mean (SD)	6.92 (3.49)	7.55 (3.04)
Number of PRN drugs	*n* = 106	*n* = 107
Mean (SD)	4.04 (2.74)	4.72 (2.89)

^a^ A lower *n* is specified in case of missing cases. ^b^ General NH wards often dedicated to people with somatic diseases who need continuous assistance. ^c^ NH ward with a higher resident:staff ratio, often dedicated to people with a severe degree of dementia and neuropsychiatric symptoms. ^d^ NPI-subsyndromes are calculated as the sum of the following items: NPI-Agitation = Agitation + Disinhibition + Irritability; NPI-Psychosis = Delusions + Hallucinations; NPI-Affective = Depression + Anxiety. CDR, Clinical Dementia Rating scale; CSDD, Cornell Scale for Depression in Dementia; GAI, Geriatric Anxiety Inventory; GMHR, General Medical Health Rating Scale; MADRS, Montgomery and Asberg Depression Rating Scale; MoCA, Montreal Cognitive Assessment; NPI, Neuropsychiatric Inventory; PRN, pro re nata; PSMS, Physical Self-Maintenance Scale; QUALID, Quality of Life in Late-Stage Dementia; SD, standard deviation; TUG, Timed “Up and Go” test.

**Table 3 pharmacy-10-00032-t003:** Analyses of primary outcome ^a^: Difference in change in QoL assessed with QUALID, baseline to 12 weeks.

	Control NHs	Intervention NHs
Baseline		
*n*	97	106
Mean (SD)	21.31 (6.72)	23.27 (8.03)
Week 12		
*n*	84	95
Mean (SD)	22.74 (7.64)	23.11 (8.72)
Mean change (95% CI)	−1.69 (−3.00; −0.38)	−0.18 (−1.43; 1.07)
Mean difference in change (95% CI) *p*-value	−1.51 (−3.30; 0.28) 0.101	

^a^ Mean change in QUALID score within groups and mean difference in change between iNHs and cNHs derived from results of a linear mixed model: QoL, quality of life; QUALID, Quality of Life in Late-Stage Dementia; CI, confidence interval; SD, standard deviation.

**Table 4 pharmacy-10-00032-t004:** Analyses of secondary outcomes ^a^: difference in change in clinical outcomes from baseline to Week 8/Week 12.

	Control NHs		Intervention NHs	
**QUALID**	*n*	Mean (SD)	*n*	Mean (SD)
Baseline	97	21.31 (6.72)	106	23.27 (8.03)
Week 8	89	22.45 (7.96)	97	24.03 (8.83)
Week 12	84	22.74 (7.65)	95	23.11 (8.72)
Mean change (95% CI)				
Baseline to Week 8	−1.26 (−2.36; −0.16)		−1.14 (−2.21; −0.07)	
Baseline to Week 12	−1.75 (−2.89; −0.61)		−0.21 (−1.30; 0.88)	
Difference in change	Mean (95% CI)		*p*-value	
Baseline to Week 8	−0.12 (−1.62; 1.38)		0.876	
Baseline to Week 12	−1.54 (−3.08; 0.01)		0.052	
**CSDD**	*n*	Mean (SD)	*n*	Mean (SD)
Baseline	94	6.50 (5.84)	87	7.46 (5.99)
Week 8	86	7.38 (6.19)	72	7.60 (6.91)
Week 12	77	6.49 (5.75)	60	5.80 (5.39)
Mean change (95% CI)				
Baseline to Week 8	−1.09 (−1.96; −0.22)		−0.05 (−1.02; 0.91)	
Baseline to Week 12	−0.73 (−1.66; 0.20)		1.86 (0.82; 2.90)	
Difference in change	Mean (95% CI)		*p*-value	
Baseline to Week 8	−1.03 (−2.29; 0.23)		0.109	
Baseline to Week 12	−2.59 (−3.95; −1.23)		<0.001	
**MADRS**	*n*	Mean (SD)	*n*	Mean (SD)
Baseline	78	9.03 (7.80)	45	7.47 (6.67)
Week 8	66	10.59 (8.17)	22	7.27 (5.18)
Week 12	65	10.05 (7.83)	16	7.88 (6.62)
Mean change (95% CI)				
Baseline to Week	−1.81 (−3.06; −0.56)		0.17 (−1.90; 2.23)	
Baseline to Week 12	−0.98 (−2.34; 0.38)		−0.10 (−2.62; 2.41)	
Difference in change	Mean (95% CI)		*p*-value	
Baseline to Week 8	−1.98 (−4.36; 0.40)		0.106	
Baseline to Week 12	−0.88 (−3.69; 1.94)		0.542	
**NPI-Agitation subsyndrome ^b^ **	*n*	Mean (SD)	*n*	Mean (SD)
Baseline	107	5.26 (8.38)	104	8.20 (9.48)
Week 8	98	6.70 (9.52)	92	8.64 (9.68)
Week 12	92	6.27 (9.06)	85	8.73 (10.21)
Mean change (95% CI)				
Baseline to Week 8	−1.22 (−2.57; 0.14)		−0.41 (−1.83; 1.01)	
Baseline to Week 12	−1.12 (−2.53; 0.29)		−0.46 (−1.93; 1.02)	
Difference in change	Mean (95% CI)		*p*-value	
Baseline to Week 8	−0.81 (−2.72; 1.11)		0.409	
Baseline to Week 12	−0.66 (−2.65; 1.32)		0.514	
**NPI-Psychosis subsyndrome ^b^ **	*n*	Mean (SD)	*n*	Mean (SD)
Baseline	101	1.93 (3.72)	102	3.51 (4.73)
Week 8	92	1.95 (3.45)	90	4.07 (5.88)
Week 12	85	1.85 (3.75)	81	4.30 (6.17)
Mean change (95% CI)				
Baseline to Week 8	−0.20 (−0.92; 0.51)		−0.55 (−1.28; 0.19)	
Baseline to Week 12	−0.25 (−0.99; 0.50)		−0.57 (−1.34; 0.20)	
Difference in change	Mean (95% CI)		*p*-value	
Baseline to Week 8	0.35 (−0.65; 1.35)		0.497	
Baseline to Week 12	0.32 (−0.73; 1.37)		0.548	
**NPI-Affective subsyndrome ^b^ **	*n*	Mean (SD)	*n*	Mean (SD)
Baseline	107	3.58 (5.46)	101	4.15 (5.42)
Week 8	96	4.94 (6.78)	90	4.76 (6.48)
Week 12	90	4.41 (6.12)	84	5.04 (7.04)
Mean change (95% CI)				
Baseline to Week 8	−1.19 (−2.16; −0.23)		−0.67 (−1.67; 0.32)	
Baseline to Week 12	−0.95 (−1.93; 0.04)		−0.86 (−1.88; 0.16)	
Difference in change	Mean (95% CI)		*p*-value	
Baseline to Week 8	−0.52 (−1.90; 0.86)		0.459	
Baseline to Week 12	−0.09 (−1.50; 1.33)		0.907	
**NPI-Total score**	*n*	Mean (SD)	*n*	Mean (SD)
Baseline	107	17.10 (19.10)	104	21.92 (21.30)
Week 8	98	20.11 (21.73)	92	23.79 (25.45)
Week 12	99	16.61 (19.25)	91	23.33 (27.45)
Mean change (95% CI)				
Baseline to Week 8	−2.85 (−5.90; 0.20)		−2.22 (−5.39; 0.96)	
Baseline to Week 12	0.48 (−2.59; 3.54)		−1.75 (−4.95; 1.45)	
Difference in change	Mean (95% CI)		*p*-value	
Baseline to Week 8	−0.63 (−4.98; 3.71)		0.775	
Baseline to Week 12	2.22 (−2.15; 6.59)		0.319	
**NPI-Caregiver**	*n*	Mean (SD)	*n*	Mean (SD)
Baseline	107	6.92 (8.50)	104	9.48 (10.49)
Week 8	98	7.73 (8.31)	92	9.57 (11.26)
Week 12	92	7.11 (8.49)	85	9.88 (12.05)
Mean change (95% CI)				
Baseline to Week 8	−0.79 (−1.97; 0.38)		−0.16 (−1.41; 1.08)	
Baseline to Week 12	−0.48 (−1.71; 0.76)		−0.19 (−1.49; 1.11)	
Difference in change	Mean (95% CI)		*p*-value	
Baseline to Week 8	−0.63 (−2.28; 1.02)		0.454	
Baseline to Week 12	−0.29 (−2.01; 1.43)		0.744	
**MoCA**	*n*	Mean (SD)	*n*	Mean (SD)
Baseline	79	10.66 (6.97)	73	7.08 (6.44)
Week 8	67	10.48 (6.66)	44	7.43 (6.33)
Week 12	62	10.58 (6.90)	37	7.62 (7.03)
Mean change (95% CI)				
Baseline to Week 8	0.61 (−0.37; 1.60)		0.66 (−0.55; 1.86)	
Baseline to Week 12	0.62 (−0.43; 1.67)		0.26 (−1.06; 1.58)	
Difference in change	Mean (95% CI)		*p*-value	
Baseline to Week 8	−0.05 (−1.55; 1.46)		0.953	
Baseline to Week 12	0.36 (−1.28; 1.99)		0.671	
**GAI**	*n*	Mean (SD)	*n*	Mean (SD)
Baseline	81	5.58 (5.70)	56	5.00 (5.32)
Week 8	65	5.95 (6.20)	26	3.38 (3.85)
Week 12	65	5.91 (6.20)	27	3.07 (3.09)
Mean change (95% CI)				
Baseline to Week 8	−0.78 (−1.69; 0.13)		0.91 (−0.49; 2.32)	
Baseline to Week 12	−0.35 (−1.26; 0.55)		1.27 (−0.11; 2.64)	
Difference in change	Mean (95% CI)		*p*-value	
Baseline to Week 8	−1.69 (−3.37; −0.01)		0.049	
Baseline to Week 12	−1.62 (−3.27; 0.03)		0.056	
**PSMS**	*n*	Mean (SD)	*n*	Mean (SD)
Baseline	109	1.06 (1.31)	108	1.16 (1.29)
Week 8	101	1.14 (1.52)	98	1.19 (1.26)
Week 12	98	1.03 (1.38)	95	1.02 (1.17)
Mean change (95% CI)				
Baseline to Week 8	−0.04 (−0.17; 0.09)		0.00 (−0.14; 0.13)	
Baseline to Week 12	0.04 (−0.10; 0.17)		0.11 (−0.03; 0.25)	
Difference in change	Mean (95% CI)		*p*-value	
Baseline to Week 8	−0.04 (−0.22; 0.15)		0.710	
Baseline to Week 12	−0.07 (−0.26; 0.23)		0.444	
**Charlson Comorbidity Index**	*n*	Mean (SD)	*n*	Mean (SD)
Baseline	108	2.54 (1.96)	101	2.57 (1.68)
Week 8	98	2.48 (1.84)	96	2.52 (1.65)
Week 12	94	2.50 (1.79)	93	2.57 (1.78)
Mean change (95% CI)				
Baseline to Week 8	0.04 (−0.09; 0.16)		0.04 (−0.08; 0.16)	
Baseline to Week 12	0.08 (−0.04; 0.20)		−0.04 (−0.16; 0.08)	
Difference in change	Mean (95% CI)		*p*-value	
Baseline to Week 8	−0.00 (−0.18; 0.17)		0.984	
Baseline to Week 12	0.12 (−0.05; 0.30)		0.169	
**TUG**	*n*	Mean (SD)	*n*	Mean (SD)
Baseline	40	26.81 (16.67)	36	27.52 (20.36)
Week 8	25	64.84 (110.98)	20	36.22 (25.52)
Week 12	24	83.01 (136.12)	20	40.56 (26.94)
Mean change (95% CI)				
Baseline to Week 8	−35.95 (−66.04; −5.85)		−9.42 (−41.98; 23.14)	
Baseline to Week 12	−52.98 (−87.12; −18.83)		−17.10 (−53.39; 19.19)	
Difference in change	Mean (95% CI)		*p*-value	
Baseline to Week 8	−26.53 (−69.52; 16.46)		0.229	
Baseline to Week 12	−35.88 (−83.38; 11.63)		0.141	
**CDR**	*n* (%)		*n* (%)	
Baseline				
No/questionable cognitive impairment ^c^	11 (10.7)		8 (7.7)	
Mild cognitive impairment	30 (29.1)		20 (19.2)	
Moderate cognitive impairment	28 (27.2)		32 (30.8)	
Severe cognitive impairment	34 (33.0)		44 (42.3)	
Week 8				
No/questionable cognitive impairment ^c^	12 (12.6)		4 (4.3)	
Mild cognitive impairment	24 (25.3)		8 (8.5)	
Moderate cognitive impairment	28 (29.5)		34 (36.2)	
Severe cognitive impairment	31 (32.6)		48 (51.1)	
Week 12				
No/questionable cognitive impairment ^c^	10 (11.1)		4 (4.3)	
Mild cognitive impairment	23 (25.6)		10 (10.9)	
Moderate cognitive impairment	26 (28.9)		28 (30.4)	
Severe cognitive impairment	31 (34.4)		50 (54.3)	
Odds of change (95% CI)				
Baseline to Week 8	0.97 (0.52; 1.83)		0.27 (0.14; 0.53)	
Baseline to Week 12	0.68 (0.35; 1.30)		0.29 (0.14; 0.57)	
Difference in change	OR (95% CI)		*p*-value	
Baseline to Week 8	0.28 (0.11; 0.70)		0.007	
Baseline to Week 12	0.42 (0.16; 1.09)		0.076	
**GMHR**	*n* (%)		*n* (%)	
Baseline				
Poor/Fair ^c^	44 (41.5)		61 (61.6)	
Good	37 (34.9)		19 (19.2)	
Excellent	25 (23.6)		19 (19.2)	
Week 8				
Poor/Fair ^c^	43 (43.4)		57 (60.0)	
Good	36 (36.4)		20 (21.1)	
Excellent	20 (20.2)		18 (18.9)	
Week 12				
Poor/Fair ^c^	41 (42.7)		55 (60.4)	
Good	41 (42.7)		17 (18.7)	
Excellent	14 (14.6)		19 (20.9)	
Odds of change (95% CI)				
Baseline to Week 8	1.22 (0.60; 2.44)		0.80 (0.35; 1.79)	
Baseline to Week 12	1.57 (0.77; 3.20)		0.96 (0.42; 2.18)	
Difference in change	OR (95% CI)		*p*-value	
Baseline to Week 8	0.66 (0.22; 1.91)		0.440	
Baseline to Week 12	0.61 (0.21; 1.81)		0.375	

^a^ A linear mixed model is used for continuous variables. A generalized linear mixed model is used for categorical variables. ^b^ NPI-subsyndromes are calculated as the sum of the following items: NPI-Agitation = Agitation + Disinhibition + Irritability; NPI-Psychosis = Delusions + Hallucinations; NPI-Affective = Depression + Anxiety. ^c^ Categories put together due to low *n* otherwise. CI, confidence interval; CDR, Clinical Dementia Rating scale; CSDD, Cornell Scale for Depression in Dementia; GAI, Geriatric Anxiety Inventory; GMHR, General Medical Health Rating Scale; MADRS, Montgomery and Asberg Depression Rating Scale; MoCA, Montreal Cognitive Assessment; NPI, Neuropsychiatric Inventory; OR, odds ratio; PSMS, Physical Self-Maintenance Scale; QUALID, Quality of Life in Late-Stage Dementia; SD, standard deviation; TUG, timed “Up and Go” test.

**Table 5 pharmacy-10-00032-t005:** Analyses of secondary outcomes ^a^: difference in change in medication prescriptions from baseline to Week 8/Week 12.

	Control NHs		Intervention NHs	
**Total number of daily drugs**	*n*	Mean (SD)	*n*	Mean (SD)
Baseline	109	6.92 (3.49)	108	7.55 (3.04)
Week 8	102	6.73 (3.69)	99	7.14 (3.00)
Week 12	99	6.65 (3.54)	96	7.18 (3.16)
Mean change (95% CI)				
Baseline to Week 8	0.16 (−0.08; 0.39)		0.56 (0.32; 0.81)	
Baseline to Week 12	0.30 (0.01; 0.58)		0.44 (0.16; 0.73)	
Difference in change	Mean (95% CI)		*p*-value	
Baseline to Week 8	−0.41 (−0.75; −0.06)		0.023	
Baseline to Week 12	−0.15 (−0.58; 0.29)		0.504	
**Total number of PRN drugs**	*n*	Mean (SD)	*n*	Mean (SD)
Baseline	106	4.04 (2.74)	107	4.72 (2.89)
Week 8	96	4.42 (2.69)	97	4.48 (3.13)
Week 12	88	4.43 (2.78)	91	4.30 (3.12)
Mean change (95% CI)				
Baseline to Week 8	−0.26 (−0.56; 0.03)		0.11 (−0.18; 0.41)	
Baseline to Week 12	−0.25 (−0.60; 0.09)		0.09 (−0.26; 0.43)	
Difference in change	Mean (95% CI)		*p*-value	
Baseline to Week 8	−0.38 (−0.80; 0.065)		0.083	
Baseline to Week 12	−0.34 (−0.86; 0.17)		0.189	
**Antidepressants**	*n*	*n* (%)	*n*	*n* (%)
Baseline	109	37 (33.9)	108	36 (33.3)
Week 8	102	35 (34,3)	99	30 (30.3)
Week 12	99	35 (35.4)	96	29 (30.2)
Odds for change (95% CI)				
Baseline to Week 8	1.00 (0.43; 2.33)		0.75 (0.34; 1.68)	
Baseline to Week 12	1.04 (0.44; 2.42)		0.77 (0.34; 1.74)	
Odds for difference in change	OR (95% CI)		*p*-value	
Baseline to Week 8	0.75 (0.23; 2.40)		0.626	
Baseline to Week 12	0.74 (0.23; 2.41)		0.623	
**Antipsychotics**	*n*	*n* (%)	*n*	*n* (%)
Baseline	109	17 (15.6)	108	29 (26.9)
Week 8	102	14 (13.7)	99	25 (25.3)
Week 12	99	13 (13.1)	96	25 (26.0)
Odds for change (95% CI)				
Baseline to Week 8	0.70 (0.25; 1.98)		0.86 (0.37; 1.98)	
Baseline to Week 12	0.67 (0.23; 1.93)		0.91 (0.39; 2.10)	
Odds for difference in change	OR (95% CI)		*p*-value	
Baseline to Week 8	1.23 (0.32; 4.65)		0.765	
Baseline to Week 12	1.36 (0.35; 5.27)		0.654	
**Sedatives and hypnotics**	*n*	*n* (%)	*n*	*n* (%)
Baseline	109	30 (27.5)	108	22 (20.4)
Week 8	102	26 (25.5)	99	21 (21.2)
Week 12	99	24 (24.2)	96	18 (18.8)
Odds for change (95% CI)				
Baseline to Week 8	0.81 (0.35; 1.90)		1.09 (0.43; 2.73)	
Baseline to Week 12	0.80 (0.34; 1.89)		0.84 (0.32; 2.17)	
Odds for difference in change	OR (95% CI)		*p*-value	
Baseline to Week 8	1.33 (0.38; 4.67)		0.652	
Baseline to Week 12	1.05 (0.29; 3.81)		0.942	
**Anxiolytics**	*n*	*n* (%)	*n*	*n* (%)
Baseline	109	22 (20.2)	108	14 (13.0)
Week 8	102	20 (19.6)	99	12 (12.1)
Week 12	99	19 (19.2)	96	11 (11.5)
Odds for change (95% CI)				
Baseline to Week 8	0.94 (0.35; 2.51)		0.83 (0.27; 2.57)	
Baseline to Week 12	0.85 (0.31; 2.28)		0.71 (0.22; 2.25)	
Odds for difference in change	OR (95% CI)		*p*-value	
Baseline to Week 8	0.89 (0.20; 3.93)		0.874	
Baseline to Week 12	0.84 (0.18; 3.84)		0.822	
**Antidementia drugs**	*n*	*n* (%)	*n*	*n* (%)
Baseline	109	9 (8.3)	108	34 (31.5)
Week 8	102	10 (9.8)	99	29 (29.3)
Week 12	99	11 (11.1)	96	29 (30.2)
Odds for change (95% CI)				
Baseline to Week 8	1.30 (0.36; 4.74)		0.80 (0.34; 1.89)	
Baseline to Week 12	1.64 (0.46; 5.86)		0.83 (0.35; 1.95)	
Odds for difference in change	OR (95% CI)		*p*-value	
Baseline to Week 8	0.62 (0.13; 2.90)		0.541	
Baseline to Week 12	0.51 (0.11; 2.35)		0.385	

^a^ A linear mixed model is used for continuous variables. A generalized linear mixed model is used for categorical variables. CI, confidence interval; OR, odds ratio; PRN, pro re nata; SD standard deviation.

## Data Availability

The data that support the findings of this study are available on request from the corresponding author, but restrictions apply to the availability of these data. The data are not publicly available due to privacy or ethical restrictions. However, data may be available from the authors upon reasonable request and with permission of the Regional Committee for Medical and Health Research Ethics.

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
