# Peer review of "The Effect of the NorGeP–NH on Quality of Life and Drug Prescriptions in Norwegian Nursing Homes: A Randomized Controlled Trial"

_pharmacy, 2022, doi:10.3390/pharmacy10010032_

Round 1

Reviewer 1 Report

Dear authors,

Thank you very much for this interesting draft about using the NorGeP-NH in nursing home residents. The manuscript is very well written, the English language perfect to me as a non-native speaker. The introduction, methods sections and results are fine and clear, the limitations (12-weeks to follow up etc.) are pointed out thoroughly. 

Only one recently published trial (HIOPP-3-iTBX) from Germany is missing, a RCT aiming to reduce a PIM/antipsychotics (Junius-Walker U, Krause O, et al, 2021). 

https://www.aerzteblatt.de/int/archive/article/221644

This study should be added in the introduction, e.g. on page 2, line 68, to have all the RCTs to this topic up to date.  

Reviewer 2 Report

The authors conducted an important study for clinical practice, since polypharmacy in patients treated at home significantly increases the risk of adverse drug reactions. The goals of this are sufficiently reasoned by the authors. The article is well structured, easy to read and is of undoubted scientific and clinical interest.

I think that the manuscript needs a little technical correction: corrections of the affiliation of the co-authors, all tables and links to the standard of the journal.

Reviewer 3 Report

Abstract, line 30: Your conclusion can not be decisive on the effectivness of NorGep. Please replace 'is scarce' with 'may be scarce'. Also, add this sentence: Further studies about the effectivness of NorGep in other healthcare contexts and settings are recommended. 

Introduction lines 72-74: Your claim can not be always true! You know that medicines management is a team work and all involved in healthcare have responsibilies toward it. Nurse collect data on signs and symptoms, as the prescriber is not 'alltid' available in the patient's bedside. Pharmacist helps with medication checking and review and collectively, nurse and pharmacist, give reports and suggestions to prescriber to make a decision. Absolutely this is the prescriber's role to make the final decision, but the medication review process needs a team collaboration. Please revise your sentences to give a scientific conclusion to international readers.

Methods: With single-blinded, who has been blinded exactly. You mentioned that you could not blind the participants given the intervention identity. If they have not been taken blinded throughout the research, why do you call this study single-blinded?

Line 110, what do you mean by healthcare personnel?

Line 144, you stated that healthcare staff collected data after receiving education about it. How can you be sure that 'inter-rater' differences have not led to bias in the data collection process? Have you done a pilot-test or calculated inter-rater reliability to ensure that differences in individual data collectors have not led to bias?

The whole methods' section can be summarised and reflected using a figure.

Table 1, Who has collected data using all these data collection tools?

Lines 176-181 include data collection elements that do not match the tools in Table 2, such as PRN. How these data have been collected? 

Lines 209, 210, please be elaborative on the criteria for stratifying. Was the type of disease etc?

Line 253, what do you mean by some 'NHs could not include more than a predetermined number of residents'?

Line 257, How many healthcare personnel were involved in the data collection?

Did you inform physicians in the cNH about your intervention that could lead to their 'maturation' and probable 'medication review' by themselves without noticing the researcher?

Results: Table 2, did you assess the groups in terms of background data to ensure of their comparability/homogeneity before the intervention?  The table has no such an information. 

Have these patients in i/cNHs ben comparable in terms of types of taken medications and diseases leading to taking medications in the baseline data collection phase?

Lines 362-363, thus you can not call your research single-blinded, because they have not been blinded throughout the research.

Your research conclusion should include suggstions for future research, policy making and clinical practice.

Round 2

Reviewer 3 Report

Nothing more.